# Children First, a Debate on the Restrictions to Tackle COVID-19

**DOI:** 10.3390/children10020211

**Published:** 2023-01-25

**Authors:** Sergio Verd

**Affiliations:** 1Pediatric Unit, La Vileta Surgery, Department of Primary Care, Baleares Health Authority, Matamusinos Street, 07013 Palma de Mallorca, Spain; drsverd@gmail.com; 2Balearic Institute of Medical Research (IdISBa), Valldemossa Rd., 07120 Palma de Mallorca, Spain; 3Clinical Ethics Committee, Department of Primary Care, Escola Graduada Street, 07003 Palma de Mallorca, Spain

**Keywords:** COVID-19, child, codes of ethics, clinical ethics, preventive measures, cost benefit analyses

## Abstract

Sometimes, when a public health disaster strikes, mandatory freedom-limiting restrictions must be enforced in order to save lives. During the first waves of the COVID-19 pandemic, the customary and necessary exchange of ideas in academia drastically changed in most countries, and the absence of debate on the restrictions enforced became evident. Now that the pandemic seems to be drawing to an end, the aim of this article is to spark clinical and public debate on the ethical issues concerning pediatric COVID-19 mandates in an attempt to analyze what happened. With theoretical reflection, and not empirical inquiry, we address the mitigation measures which proved detrimental to children despite being beneficial to other segments of the population. We focus on three key points: (i) the sacrifice of fundamental children’s rights for the greater good, (ii) the feasibility of cost–benefit analyses to make public health decisions and restrictions which affect children, and (iii) to analyze the impediments to allowing children’s voices to be heard concerning their medical treatment.

## 1. Introduction

As a gesture of good will to our patients and admittance of human fallibility, we want to open a clinical and public debate on the ethical issues concerning pediatric COVID-19. At the start of the most recent health emergency, while scientists around the world acted extraordinarily fast to investigate SARS-CoV-2 and COVID-19, policymakers worldwide made undisputable decisions regarding fundamental public values (invoking but not debating fundamental values that already seemed clear). At the same time, the limited dissent to restrictions by state authorities has been striking, particularly among defenders of human rights. Despite pandemic plans prior to 2020 emphasizing the importance of using the least restrictive measures possible, populations generally accepted lockdowns as a necessary instrument of Public Health [1]. Given the complexity of most clinical ethical issues, more than common-sense belief was needed for unanimity on the fairest distribution of the damage inflicted on different population groups by COVID-19 prevention. Now that the pandemic seems to be drawing to an end, the blame is often put on the lack of a perfect balance between conflicting co-existing principles or priorities. We hope this Special Issue of Children will attract different points of view on the ethical questions that have emerged with pediatric COVID-19. Recent guidelines of the Council of Europe stress that there is hardly any justification for the prior censorship of specific topics [2]. Similarly, medical professionals, and members of the general public, have the right to scrutinize how far the authorities’ responses to the crisis should extend and how long they should remain in place. As a matter of public health, this should not be too slow, insufficient, or too risky for researchers’ careers.

Under emergency measures, academic freedom is crucial in preventing panic and fostering people’s cooperation with the demands of the situation. At the beginning of the COVID-19 crisis, an increase in confidence in research was recorded, but trust in researchers gradually decreased as the pandemic continued. In April 2020, most British interviewees were convinced that the pandemic made them more likely to listen to advice from qualified experts. However, the public’s faith in research fell from 73% in April 2020 to 60% in November 2020, according to the German Wissenschaftsbarometer; and the Eurobarometer has shown a similar trend among European respondents [3]. At the same time, the vibrant exchange in academia underwent drastic changes in most countries, and the atmosphere of self-censorship threatened the open environment needed for research to thrive. In addition, academic freedom is not decoupled from all social or legal constraints people face; where political polarization is low, academic freedom is high, and vice versa [4]. Not only academic freedom was constrained; individual liberties were also restricted in unprecedented ways in many countries around the world during the COVID-19 pandemic, whether democratic or authoritarian [5].

Crises require particular caution in introducing conversation stoppers. When debating controversial ethical tissues, one can hear the pernicious claim, “That is just an opinion.” The opinion “label” ends a dispute by reducing it to a matter of taste. This conclusion is especially troubling in ethical debates. It cannot be forgotten that these are vital debates because they not only matter to us; they are about what to do. In these cases, people cannot refrain from standing up for their convictions even if they may be interpreted as personal opinions as long as people can defend that they are well-informed and well-reasoned. By extension, we propose that we abandon any potential infallible sources of evidence and that the debate must continue, however hard that may be.

Kant distinguished between knowing and thinking [4,6]. While knowledge is nourished by evidence, thinking is not satisfied with the evidence. It goes further, it progresses to elaborate a more complete interpretation of things; it is inherent to our historical and biological peculiarity.

In any case, examining every angle of the ethical issues that arise from COVID-19 and its effect on children is “strictly required by the exigencies of the situation” and should be made explicit. The goal of requesting papers on this very sensitive social topic is to reduce skepticism about the integrity behind pediatric COVID-19 mandates. There is a risk that society will become more of an “allegiant society”, a term used by political scientists and characterized by being focused on security and having deference to authority [7], in contrast to an “assertive society”, which values civic engagement and authority skepticism. Societies may opt to follow the allegiant path or not, but lacking a clear knowledge of the principles at stake and their willingness to pay a price makes it difficult for them to strike a balance between freedom and security.

This ethical examination does not conclude with a thorough comprehension of the facts and a clear understanding of the consequences of our values, but it is an excellent place to start. To begin with, we would like to focus on a point that adds an extra layer of complexity. We will refer to public health campaigns of mitigation measures that are detrimental to children yet beneficial to other segments of the population during sanitary emergencies.

## 2. Methods

According to the taxonomy for clinical ethics research [8], this study used theoretical rather than empirical methods. The use of qualitative research in bioethics is growing in popularity; this approach involves scholars working at the intersection of clinical practice and the humanities, dedicated to addressing normative questions that require ‘slow-reading’ through profound theoretical reflection, in order to formulate ethical guidance that is in-tune with the nuances of the topic.

Conducting quality studies in clinical ethics does not come without challenges. There are different criteria for judging quality, and reporting results [9]. Finally, this approach is a complement and not a substitute for empirical research in the field. Any methodological standpoint alone is incomplete. Since empirical and theoretical studies are synergistic, their combination offers research opportunities that neither could achieve separately [10].

## 3. The Strong Normative Force of Children’s Rights

Mandatory orders during natural disasters save lives but limit human rights. No exception has been made during the COVID-19 pandemic. The principal tactic used by policymakers to safeguard public health during the pandemic has been urging adults and children to change their behavior. However, adult participants in empirical ethical research during the 2020 COVID-19 wave were reluctant to violate personal rights to create the best overall utilitarian outcomes [11]. In summary, people are not willing to sacrifice individual rights for the sake of the common good. This discovery has significant ramifications for policy choices where these trade-offs may be present. To the best of our knowledge, no similar surveys were conducted in which children’s moral judgment was taken into account.

Towards the end of her life, Anna Freud was a visiting professor at Yale, not in psychology, but at the law school, and there, her goal was “for children to have lives that would not be oppressed, restricted, impoverished, or damaged by an uncaring adult world.”

Until the middle of the 19^th^ century, nobody thought of giving special protection to children. Since then, laws have started to protect children in their workplace, subsequently expanded to the right of children to be educated, and finally to the medical or social fields. One highlight in the long and complex history of advancing the rights of children has been that 196 out of 197 states have ratified the United Nations Convention on the Right of the Children (UNCRC). Under the terms of the UNCRC, governments are required to acknowledge that every child has basic fundamental rights. However, the various rights of children have proven less than successful in the face of the disturbing reality of many children’s lives.

Children’s rights depend on children’s cognitive development and have been classified as protection (nurturance or welfare) rights and free choice (self-determination) rights [12]. Nurturance rights are the child’s right to the provision of education, housing, and healthcare. Self-determination is the right of children to choose or to make their own decisions. Some countries consider welfare rights to be more beneficial to children, while others support self-determination. In any case, exercising nurturance rights is used to leave children without self-determination rights and vice versa. In the view of protectionists, children need protection, while from the point of view of liberationists, they should have an active role in society. The solution of pragmatists is somewhere in between; they recognize that children cannot manage successfully in the world, but they also claim greater flexibility according to the child’s emerging sound judgment [12].

In other words, the law has various distinctions between how it treats children and adults. There is much more literature on children’s own set of rights than on the need to apply to children without restrictions or delay all the rights enjoyed by adults. There is some resistance to extending to children the rights we adults demand ourselves. There is also a trend to believe that children should have their unique set of rights and that it would be absurd to grant children the same rights that adults do. This line of thinking might be harmful to children, just as it was harmful to historically oppressed groups that were treated differently from the average person in law. Children might have some needs that other people do not. There are some groups of adults with special needs, but they enjoy the same rights as everyone else. In fact, the possibility of groups of people having different sets of interests does not inapt the idea of treating them equally at a fundamental level.

The debate on whether to treat them the same, as simple individuals to whom all laws should be applied, or differently, currently tends to be on the side of treating them differently, but the former alternative can be important for the field of “children studies” in the future. We should decide when and to what extent it is morally acceptable to not use a principle of equality between adults and children.

## 4. A Dignity Frame to Secure the Well-Being of Children

We should fully reflect on the justification of the adverse effects suffered by a generation of children, and particularly the most disadvantaged, by public health restrictions that affect children but benefit other groups of the population rather than benefiting them. The ability to resist the magic of very seductive cost-effectiveness analysis (CEA) is one of the most important qualities a policymaker should possess. CEA is an appealing tool for evaluating health-related interventions. In order for cost–benefit calculations to be performed, all costs and benefits must be expressed in a common measure, including things that are “not for sale”. In numerous ways, the mere declaration that something is not for sale increases and protects that thing’s value. We render a final determination of its unique value. We cannot place prices on not-market things [13].

Cost inclusion in healthcare policy has been extensively studied from an ethical perspective; if researchers do not take a society’s ideals into account when allocating resources in a way that does not represent societal values, communities will pay a high price [14]. There are approaches to bridge the gap between CEA results and society preferences by incorporating distributive justice and equitable ideals into the models (i.e., age or illness severity) [15]. Unfortunately, weighing one group will inevitably detract from another when accounting for equity in a utilitarian model. The emphasis is placed on the assumption that the presence of reliable evidence ensures that better decisions will be made. The appeal to formulas satisfies the wish that we have for objective methods for the resolution of the controversy. In theory, this approach seeks to resolve moral deliberation questions by transforming them into problems of measurement. This method narrows the deliberative process to a decision based on a calculation of superficial and broad projected outcomes rather than on deep individual experience [16]. In this accounting, the utility of mandates against COVID-19 on healthy children can be expressed as numbers of quality-adjusted life-years gained in the whole of society. However, these calculations will not answer the “ought” question of bioethics. Decisions involve values, and science cannot tell which values we should put weight on; these are ethical decisions. The crucial question remains open: how did this society arrive to a point where children were subjected to restrictions or even interventions, not for the purpose of benefiting them, but in order to pursue adults’ protection?

Among others, the potential consequences for children’s well-being and the behavior of restricting freedom merit analysis. A large amount of research shows that autonomy plays a crucial role in health. People who feel that their freedom has been thwarted are at greater risk for depression, anxiety, suicidality, and also tend to suffer worse physical health [17]. The detrimental effects of restricting freedom have been documented across many different cultural contexts. Notably, children who have less control over how and with whom they spend their time are less happy and less successful in school [18]. Previous research on suspending constitutional rights from SARS, and Ebola pandemics, showed its profound impact on exhaustion, anxiety, and posttraumatic stress symptoms [19]. Similarly, psychological distress linked to COVID-19 quarantine have begun to emerge. An obstacle to balancing the cost–benefit of restricting children’s freedom is that freedom is measured for its contributions to sustainable development goals and for its potential to fuel productivity, but is also a sacred value, a moral good in itself.

Before the COVID-19 pandemic, the International Code of Medical Ethics [20] stated that the individual clinician is to “always exercise his/her independent judgment… and to act in the patient’s best interest when providing medical care”, and the European Charter for Children in Hospitals stated that “every child should be protected from unnecessary medical treatment” [21]. The largely individualistic focus of codes of professional ethics encourages diversity in treatments and comes into conflict with the efforts to contain the pandemic that has taken a serious toll on children and has apparently disenfranchised children. Most government organizations appeared to have forgotten their adherence to the UN Convention on the Rights of the Child during the pandemic. Children’s best interests have obviously not been prioritized during the past two years. Amidst this tension, we would like to recall that “Children first” in threatening situations has been the code of conduct in civilized societies century after century, ranging from Roman law (In dubio pro liberis) to 19^th^ century shipwrecks.

## 5. Disregarding Children’s Voices from COVID-19 Policy Discussion

Actual consent is a necessary condition to avoid using children merely as a means. However, we sense that the approach to children’s consent to the last pandemic restrictions that have been taken not in their own interests, but in the interests of others has been notoriously brief.

A comprehensive review of recent studies on children’s views and priorities to inform the next Council of Europe Strategy for the Rights of the Child [22] emphasizes that children are acutely aware of discrimination against them on the basis of their young age; children in Europe want to be heard and have some say in decisions affecting their own care. Youngsters frequently believe that adults do not trust them, and when attempts to have their opinions heard do not produce feedback, children lose hope. This attitude has been widespread during the pandemic. In fact, the UK government has been accepting public questions for the daily COVID-19 briefing since April 2020. Surprisingly, the conditions for question submission expressly forbid queries from youngsters [23]. If a person under the age of 18 tries to submit a question, an error notice appears saying, “Sorry, you cannot submit a question.”

Most authors report that children express the desire to have their voices heard when it comes to decisions about their medical treatment [24]. Several schemata have been developed over time to structure children’s participation in consultation. However, many children believe that they have been excluded from decisions affecting their bodies [25]. The fact that the experiences of children from various locations were similar suggests that communication difficulties between children and health professionals are common.

Not only are there administrative impediments to allowing children’s voices to be heard, but there are also features of their cognitive development that make it difficult for their views to be properly understood.

One of the deontological theory’s most important tenets is the ban on using people merely as a means [6]. Among others, it influences important moral standards for medical research involving human subjects. A child must have the ability to express their dissent and to withhold their agreement to being treated merely as a means. Here, a cause for concern is not the absence of consent, but its impossibility: children cannot consent to the treatment. Given the children’s context (e.g., moral development, cognitive limitations, relatives who exert considerable influences on their daily life, and so forth), it is not a reasonable belief that children are happy with the decision that the experiment’s overwhelming benefits for large numbers of human beings outweigh their own adverse effects.

A number of theories of moral development and behavior have established that children’s moral judgment changes as they get older [26]. Children up to age 9–10 regard morality as obeying the rules that adults impose upon them; this means a morality that is formed out of being subject to some authority figures’ rules; this is the stage of heteronomous morality—morality imposed from the outside (Stage 1); profound obedience to authorities of children at moral stage 1 rules out their dissent and thus renders them unable to give consent to avoid being treated as a means. Instead, for children of 10 and over, the major motivating factor in good behavior is social approval from those closest to the child; this change is partly seen as a result of the growing importance of the peer group (Stage 2). It is doubtful that they are able to choose freely rather than being told what to do by their family or by their very significant others. Finally, some people beyond adolescence reach the stage of universal ethical principles; at this level, the emphasis is no longer on conventional standards of morality, but rather on idealized principles (Stage 3).

Surveys have reported that children divide medical care decision making into two categories: serious and small. Children are eager to participate in making small decisions about everyday events or procedures. When it comes to long term decisions, around half of children would rather leave them to their parents [24].

On account of that mentioned above, children’s simultaneous declaration of will cannot be bypassed, but there is little evidence to support that they are able to give genuine consent to altruistic behaviors.

## Data Availability

No data are associated with this article.

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
