# Peer review of "Children First, a Debate on the Restrictions to Tackle COVID-19"

_children, 2023, doi:10.3390/children10020211_

Round 1

Reviewer 1 Report

Literature review should be improved 

why did the select this method?

Author Response

Comments from Reviewer 1:

We appreciate the time that you have dedicated to providing your valuable feedback on
this manuscript.

English language and style: (x) English very difficult to understand/incomprehensible.

Response: The draft has been edited by a native English speaking doctor; all spelling and grammatical errors have been corrected.

Literature review should be improved

Response: Both you and other reviewers commented on this point. We have been able to incorporate new references. We have highlighted the changes within the manuscript.

why did the select this method?

Response: We have added a Methods section in which we explain our choice of methods.

Reviewer 2 Report

1. I am surprised by the lack of an abstract at the beginning of the article.

2. The purpose of the article has been stated. The goal of requesting papers on this very sensitive social topic is to reduce skepticism about the integrity behind pediatric COVID-19 mandates. It should have been worded better.

3. Literature review is insufficient.

4. Research hypotheses have not been put forward.

5. Studies have not been conducted.

6. The proposed text may be an introduction to the actual article.

Author Response

Comments from Reviewer 2:

We are grateful to Reviewer 2 for her/his insightful comments on our manuscript. We have done our best to reflect most of her/his suggestions.

1. I am surprised by the lack of an abstract at the beginning of the article.

Response: We agree and have taken your advice. The abstract might induce scholars to read the paper in more detail.

2. The purpose of the article has been stated. The goal of requesting papers on this very sensitive social topic is to reduce skepticism about the integrity behind pediatric COVID-19 mandates. It should have been worded better.

Response: Thank you so much for catching these confusing errors. We have expanded on the purpose of this article. We have done our best to link the aim and the content of this paper.

3. Literature review is insufficient.

Response: Both you and other reviewers commented on this point. We have been able to incorporate new references. We have highlighted the changes within the manuscript.

4. Research hypotheses have not been put forward.

Response: Thank you for this suggestion. It would have been interesting to explore this aspect. However, in the case of our study, we do not defend a hypothesis. Our aim is not to argue that a particular ethical approach must be adopted. There may be good reasons to deviate from any relevant ethical theory. However, identifying and taking into account the necessary cost of these additional ethical principles can benefit societies.

5. Studies have not been conducted.

Reviewer 2 has raised an important point here. We have added a Methods section to explain that the present study is not empirical, but a theoretical reflection. We recognize that in this way the research is partial.

6. The proposed text may be an introduction to the actual article.

While we appreciate the reviewer’s feedback, we respectfully disagree. We think this study is not only an introduction, but it makes a valuable contribution to a field of study that has been somewhat neglected.

Reviewer 3 Report

I want to thank the author for the opportunity to read his work. The viewpoint paper “Children first, a debate on the restrictions that have been put in place to tackle COVID-19. Did we fail our children?” represents an interesting contribution to various disciplines, from health information, psychology, and medicine to communication studies and social sciences, among others. The objective is well-designed in the title. The literature review is not extensive but provides the theoretical lens for the proposed debate. In short, I would suggest improving it with a critical point of view. At the conclusion or discussion level, the three parts (Children’s rights; Quantifying ethics; and, Unlocking children’s voices) are disconnected from the title/objective. In this sense, there are missing points on the two main topics: i) did we fail children, and ii) a debate on Covid 19 restrictions affecting children’s rights. I recommend linking it with the three epistemological lenses provided. As a suggestion, contextualizing or placing the debate in a jurisdiction, country, or place could make the viewpoint stronger and sharp. We believe these major changes will make the paper more focused and attractive to open a broad interdisciplinary debate on what has happened to children’s rights from an ethical viewpoint during Covid-19.         

Author Response

Comments from Reviewer 3:

We are grateful for the important points that Reviewer 3 has raised here.

Comments and Suggestions for Authors

I want to thank the author for the opportunity to read his work. The viewpoint paper “Children first, a debate on the restrictions that have been put in place to tackle COVID-19. Did we fail our children?” represents an interesting contribution to various disciplines, from health information, psychology, and medicine to communication studies and social sciences, among others.

Response: We found your comments extremely helpful and have revised accordingly.

The objective is well-designed in the title. The literature review is not extensive but provides the theoretical lens for the proposed debate. In short, I would suggest improving it with a critical point of view.

Response: Both you and other reviewers referred to the literature. We have been able to incorporate new references.

At the conclusion or discussion level, the three parts (Children’s rights; Quantifying ethics; and, Unlocking children’s voices) are disconnected from the title/objective. In this sense, there are missing points on the two main topics: i) did we fail children, and ii) a debate on Covid 19 restrictions affecting children’s rights. I recommend linking it with the three epistemological lenses provided.

Response: We agree with your suggestion. We have gone through the entire manuscript carefully and adjusted the two questions: i) did we fail children, and ii) a debate on Covid 19 restrictions affecting children’s rights, to every relevant section to clarify our meaning.

As a suggestion, contextualizing or placing the debate in a jurisdiction, country, or place could make the viewpoint stronger and sharp.

Response: In this new version of the manuscript we include clinical research on pediatric covid in our area, which has an ethical aspect.

We believe these major changes will make the paper more focused and attractive to open a broad interdisciplinary debate on what has happened to children’s rights from an ethical viewpoint during Covid-19.

Response: We agree. Thank you.

Round 2

Reviewer 1 Report

Good job

Author Response

Reviewer 1

Comments and Suggestions for Authors

Good job

RESPONSE: Thank you so much.

Reviewer 2 Report

1. The article has been revised in accordance with the reviewers' guidelines.

2. I accept the type of article as a review, although I still maintain that it can be the basis for designing new, interesting research.

3. Literature has been supplemented.

4. Titles of subchapters should be corrected so that the article does not look popular science.

Author Response

Reviewer 2

Comments and Suggestions for Authors

  1. The article has been revised in accordance with the reviewers' guidelines.

RESPONSE: Thank you.

  1. I accept the type of article as a review, although I still maintain that it can be the basis for designing new, interesting research.

RESPONSE: We will consider your suggestion for a more extensive article in the near future.

  1. Literature has been supplemented.

RESPONSE: thank you.

  1. Titles of subchapters should be corrected so that the article does not look popular science.

RESPONSE: Accordingly, the list of the new subtitles is shown below,
Introduction

Methods

The strong normative force of children’s rights.

A dignity frame to secure the well-being of children

Disregarding children’s voices from COVID-19-related policy discussions.

Reviewer 3 Report

It is important to show off our acknowledgment of the author's improvement in his manuscript based on the reviewer's report. After carefully reading, we considered this new version adjusted to the special issue and journal's scope.

Author Response

Reviewer 3

Comments and Suggestions for Authors

It is important to show off our acknowledgment of the author's improvement in his manuscript based on the reviewer's report. After carefully reading, we considered this new version adjusted to the special issue and journal's scope.

RESPONSE: Thank you so much.